# Management of Hepatocellular Carcinoma in 2024: The Multidisciplinary Paradigm in an Evolving Treatment Landscape

**DOI:** 10.3390/cancers16030666

**Published:** 2024-02-04

**Authors:** Emily Kinsey, Hannah M. Lee

**Affiliations:** 1Division of Hematology, Oncology, and Palliative Care, Department of Internal Medicine, Virginia Commonwealth University, Richmond, VA 23298, USA; 2Massey Comprehensive Cancer Center, Virginia Commonwealth University, Richmond, VA 23219, USA; 3Division of Gastroenterology, Hepatology, and Nutrition, Department of Internal Medicine, Virginia Commonwealth University, Richmond, VA 23298, USA; 4Stravitz-Sanyal Institute for Liver Disease and Metabolic Health, Virginia Commonwealth University, Richmond, VA 23298, USA; 5Hume-Lee Transplant Center, Virginia Commonwealth University, Richmond, VA 23298, USA

**Keywords:** hepatocellular carcinoma, multidisciplinary care, liver transplantation, locoregional therapies, immunotherapy, palliative care

## Abstract

**Simple Summary:**

Hepatocellular carcinoma (HCC) is a highly aggressive malignancy with global impact, especially in the context of the rising epidemic of metabolic-dysfunction-associated steatotic liver disease and alcohol-related liver disease. The treatment landscape for HCC is evolving and has changed significantly in the last few years with new treatment options for patients, while the multidisciplinary team model of care remains critical. This review aims to summarize the various treatment options for patients at all stages of HCC, highlighting the growing array of systemic therapies with multi-modal options, including the potential of combining locoregional therapies with immunotherapy across different stages of the disease. With the increasing recognition of the importance of patient-centered care, the future paradigm of HCC care includes the incorporation of non-hospice palliative care in the multidisciplinary team care model to improve patient quality of care, quality of life, and overall outcomes.

**Abstract:**

Liver cancer is the third most common cause of cancer-related deaths worldwide, and hepatocellular carcinoma (HCC) makes up the majority of liver cancer cases. Despite the stabilization of incidence rates in recent years due to effective viral hepatitis treatments, as well as improved outcomes from early detection and treatment advances, the burden of HCC is anticipated to rise again due to increasing rates of metabolic dysfunction-associated steatotic liver disease and alcohol-related liver disease. The treatment landscape is evolving and requires a multidisciplinary approach, often involving multi-modal treatments that include surgical resection, transplantation, local regional therapies, and systemic treatments. The optimal approach to the care of the HCC patient requires a multidisciplinary team involving hepatology, medical oncology, diagnostic and interventional radiology, radiation oncology, and surgery. In order to determine which approach is best, an individualized treatment plan should consider the patient’s liver function, functional status, comorbidities, cancer stage, and preferences. In this review, we provide an overview of the current treatment options and key trials that have revolutionized the management of HCC. We also discuss evolving treatment paradigms for the future.

## 1. Introduction

Liver cancer is the sixth most common cancer, but its aggressive nature and poor prognosis raise it to the third highest cause of cancer-related deaths [1]. Hepatocellular carcinoma (HCC) makes up approximately 75% of primary liver cancer cases, with a smaller proportion due to cholangiocarcinoma. The global burden of HCC is highest in Asia and sub-Saharan Africa due to the high prevalence of chronic hepatitis B (HBV) in those regions. Men are more commonly affected and have a higher mortality than women, with liver cancer being the leading cause of cancer death among men in over 20 countries [1].

Most patients develop HCC in the setting of cirrhosis, and the risk factors for HCC include HBV, hepatitis C (HCV), alcohol-related liver disease (ALD), and excess body weight and type 2 diabetes, which is often associated with metabolic-dysfunction-associated steatotic liver disease (MASLD), as well as aflatoxin exposure. The prevalence of various risk factors for HCC is region-specific, with viral causes being more common in the East and non-viral etiologies more common in the West. In Japan and Egypt, HCV is the primary driver of HCC [2,3]. In Asia and Africa, chronic HBV, a primary cause of HCC, occurs either through vertical transmission perinatally or horizontal transmission (exposure to infected blood from child to child, unsafe medical and injection practices, or unscreened blood transfusion). Due to HBV vaccination programs and hepatitis treatment options, a decline in the incidence of HCC has been noted in the East in recent years [4]. Despite this, the burden of HCC is anticipated to rise again due to the increasing rates of MASLD and ALD [5].

The treatment landscape for HCC is evolving and requires a multidisciplinary approach, with input from hepatologists, hepatobiliary/transplant surgeons, medical oncologists, diagnostic and interventional radiology, and radiation oncology. In addition, the early incorporation of a palliative care team in the treatment of HCC patients is increasingly recognized as an important component of patient-centered care and improving patient quality of care (Figure 1). This multidisciplinary approach is important as HCC management is not a “cookie cutter” process or a “one size fits all” solution. This will involve a management approach that is individualized and customized, taking into consideration multiple patient factors, including the state of the liver (cirrhosis vs. no cirrhosis) and liver function (compensated or decompensated), the size, location, and extent of the cancer, any co-morbidities, the functional status of the patient, and patient preferences. The Barcelona Clinic Live Cancer (BCLC) staging system takes many of these factors into consideration (Figure 2). Liver cancer tumor biology is also heterogeneous, with some tumors exhibiting more indolent behavior and others being more aggressive. Diagnosis and management require a collaborative team approach, with treatment customized for each patient. This personalized strategy ensures the best treatment options within the setting of their chronic liver disease to optimize outcomes. Given the complexity of care for these patients, crossing multiple disciplines, it is not surprising that outcomes are improved with discussions at a multidisciplinary liver tumor board [6,7]. Patients treated by a multidisciplinary tumor board, ideally with all specialists in a co-located clinic, are diagnosed at earlier stages, have decreased times to treatment, higher rates of therapy receipt, increased access to curative treatments, and improved overall survival. Moreover, patient satisfaction is improved [8,9,10].

Treatment can be divided between curative and non-curative approaches. For patients with localized HCC without cirrhosis or with cirrhosis but without clinically significant portal hypertension, resection is the recommended approach; however, recurrence rates are high in this setting, with a 50–70% recurrence after 5 years [11,12,13]. Liver transplantation (LT) is considered for non-resectable patients due to liver dysfunction, portal hypertension, or multi-tumor involvement. Locoregional therapies (LRTs) are part of the treatment armamentarium for patients with intermediate-stage disease but can also be used as a bridge to transplantation in early-stage disease. Patients with advanced or extrahepatic disease should be considered for the various systemic therapy options available.

Best supportive care, including consideration of the incorporation of palliative care, is the preferred option for patients with poor performance status or decline in liver function regardless of whether the treatment goals are curative or non-curative/palliative. The integration of palliative care (including non-hospice care) is particularly important given the complexity and challenges of the dual diagnosis of HCC and cirrhosis in this patient population. This patient group can bear considerable physical (ascites, variceal bleed, hepatic encephalopathy, sarcopenia, and frailty), psychosocial, and financial burdens as well as potential stigmatization. Therefore, particular emphasis is placed on both multidisciplinary and interdisciplinary care in this patient population.

The purpose of this review is to provide an overview of the current treatments of HCC with a focus on patient selection and outcomes for each treatment option. The upcoming changes in the paradigm of HCC care will also be reviewed including the advancement of systemic therapy and its use as a multi-therapy/multi-modal approach.

## 2. HCC Prevention and Surveillance

Treatment with antiviral therapy for both HBV and HCV does significantly decrease the risk for HCC in patients with or without cirrhosis [14]. HBV vaccination for the prevention of chronic HBV infection also has been shown to reduce the risk of HCC. For MASLD, other than controlling risk factors that are modifiable, such as obesity and diabetes type 2/insulin resistance, there are no clear effective HCC prevention interventions available at this time [15]. However, a recent meta-analysis of nine studies assessed the incidence and risk of HCC following bariatric surgery [16]. The pooled rate/1000 person-years was 0.05 (95% CI: 0.02–0.07) in bariatric surgery patients and 0.34 (95% CI: 0.20–0.49) in the control group, while the incidence rate ratio was 0.28 (95% CI: 0.18–0.42). In addition to providing durable weight loss, bariatric surgery may also be associated with a decreased risk of HCC.

HCC surveillance is targeted toward populations that are considered at-risk. The American Association for the Study of Liver Diseases (AASLD) Practice Guidance recommends that all patients with cirrhosis of any etiology and non-cirrhotic chronic HBV undergo surveillance every 6 months [17]. For HCV with stage 3 fibrosis or non-cirrhotic MASLD, the annual incidence of HCC is < 0.2%; thus, there is insufficient risk to warrant regular surveillance at this time for this patient group. The standard approach to HCC surveillance as recommended by the AASLD is abdominal ultrasound with AFP at 6-month intervals. If imaging with abdominal ultrasound is suboptimal, contrast-enhanced imaging with MRI is recommended.

The implementation of surveillance programs involving standardized screening protocols, recall procedures, and quality control measures is essential to decrease HCC-related deaths. HCC surveillance is associated with early diagnosis and improved survival; however, approximately 20% of cirrhotic patients only undergo semi-annual surveillance. The underuse of surveillance in clinical practice remains an ongoing challenge, particularly in patients with ALD and MASLD-related cirrhosis and patients not seen regularly by gastroenterologists [18,19].

Studies have evaluated both patient-reported barriers and primary care provider practice patterns and barriers regarding HCC surveillance. Lack of knowledge, financial limitations, scheduling difficulties, and transportation issues are some of the patient-reported barriers that are significantly associated with less frequent receipt of HCC surveillance, while primary care providers have reported misconceptions about their knowledge of surveillance [20]. Both patient-centered interventions as well as provider education are needed to improve HCC surveillance in clinical practice. Further discussion of HCC prevention and surveillance is beyond the scope of this review.

## 3. Curative Approaches

### 3.1. Resection

Patients are candidates for resection if they do not have cirrhosis or have BCLC 0/Child–Pugh (CP) A cirrhosis without portal hypertension. (Table 1). Resection with partial hepatectomy is potentially curative with a solitary tumor of any size with no evidence of gross vascular invasion. To be considered for resection, a patient needs to have an appropriate tumor location and adequate liver reserve and liver remnant. Given the relatively restrictive criteria, only about one-third of patients evaluated for resection will be able to undergo curative-intent surgery [21]. Minimally invasive techniques, involving both laparoscopic and robotic approaches, have become the standard surgical approach as they allow for more robust liver remnants, less surgical risk and complications, and faster post-surgical recovery. Portal vein embolization is an additional technique to allow more patients access to surgery. Portal vein embolization impedes blood flow to the part of the liver to be resected, and redirects portal blood flood to the non-tumor-bearing liver, thus inducing hypertrophy of the future liver remnant.

Survival rates for liver resection in well-selected patients are very encouraging, with approximately 70% at 5 years [11,13]. Even in patients with large tumors, resection still has the potential for cure, albeit with reduced survival rates compared to smaller tumors [12]. However, recurrence rates approach 50–70% at five years [11,12,13]. Most recurrences occur in the first two years, but there is a bimodal distribution with a second peak between years 4 and 5 [27]. The presence of vascular invasion and/or multifocal disease places patients at an even higher risk of recurrence; therefore, resection is not recommended in these cases.

In contrast to guidelines from the AASLD and the European Association for the Study of the Liver, the Eastern and Italian multisociety guidelines recommend consideration for resection in well-selected cirrhotic patients with good liver function who have oligo-nodular HCC (2–3 nodules). They advise that such cases receive extensive review by the multidisciplinary board and may be considered on a case-by-case basis [28]. This is based on one randomized control trial and multiple observational studies. This randomized control trial showed a longer survival rate after liver resection of patients outside of Milan criteria compared to TACE at 1 year (76% vs. 52%) and at 5 years (51% vs. 18%) [29].

In addition, for patients with macrovascular invasion (MVI), while generally considered a contraindication for resection by the AASLD and EASLD guidelines, the Eastern and Italian multisocietal guidelines would again consider liver resection in selected patients [28]. This is based on studies showing postoperative mortality rates of 3–6% and survival rates of 3 and 5 years at 17–49% and 10–30%, respectively [30,31,32,33]. The Italian society guidelines note that the site of the portal MVI with more peripheral branch involvement is associated with better prognosis, and that survival advantage after surgery compared to nonsurgical approaches has been reported only in patients with MVIs that do not extend to the portal trunk [32,34,35].

### 3.2. Transplant

For patients who are not resection candidates, LT should be considered if the tumor is within the Milan criteria or United Network for Organ Sharing (UNOS) stage T2. LT has both a high survival rate and a low recurrence rate, making it an ideal option for patients who are candidates. In addition to potentially curing the HCC, it also addresses the underlying liver disease, which is often the driver of mortality. The recurrence rates after LT are around 10%, which is much lower than with resection or ablation [25]. The Milan criteria are the most widely accepted criteria to determine if a patient should be considered for transplant for HCC and are defined by one lesion measuring between 2 cm and 5 cm or up to 3 lesions, none greater than 3 cm with no vascular involvement or extrahepatic spread. When patients within these parameters receive a transplant, their outcomes are similar to those of patients who are transplanted for non-malignant reasons [26,36]. In addition, consideration for LT requires the biomarker AFP to be <1000 ng/mL. Per UNOS policy, patients with AFP ≥ 1000 ng/mL are not eligible for MELD exception points and will require LRT with a decrease in AFP to <500 ng/mL.

If the patient is not within the Milan criteria, various LRTs can be used to downstage and bridge a patient to transplantation. The UNOS downstaging protocol (UNOS-DS), which includes patients whose total sum diameter is up to 8 cm, allows patients to gain MELD (model for end-stage liver disease) exception points if the patient can be downstaged with liver-directed therapies [37]. Most patients can be downstaged from the UNOS-DS criteria, and their survival and recurrence rates are excellent [38,39]. In a large cohort study of transplant recipients, 10-year recurrence rates were slightly higher for patients who were downstaged compared to patients who were initially within the Milan criteria (20.6% vs. 13%), but lower compared to patients who were not downstaged at all (41%) [40]. For patients who can be downstaged, the 5-year overall survival (OS) is greater with LT compared to locoregional or systemic therapies (77.5% vs. 31.2%); therefore, LT is preferred if it is an option [41].

MELD exception points for transplant are only awarded if an HCC lesion is at least 2 cm in size. For patients with cirrhosis but with a lesion < 2 cm in size, the recommendation is close observation initially. Once the tumor reaches 2 cm, and is therefore eligible for MELD exception points, the patient can undergo LRT while undergoing transplant evaluation or during the waiting period on the transplant list.

### 3.3. Ablation

Ablation alone with radiofrequency ablation (RFA) or microwave ablation (MWA) may be considered a potential curative treatment for HCC lesions up to 3 cm when LT and resection are contraindicated. With low complication rates and cost-effectiveness, ablation is associated with survival outcomes similar to resection for small tumors, making it an attractive treatment option [22,42,43]. Lesions less than 3 cm in size are ideal since the effectiveness and survival after ablation are inversely proportional to size with a significant difference in survival when using a cutoff of 3 cm [44]. The NCCN guidelines reserve ablation for patients who are not surgical candidates. The AASLD recommends ablation as an alternative option to surgery if patients have very early-stage HCC (BCLC-0)/UNOS stage T1 and transplant is not being considered [17].

Evaluation of tumor location is key for thermal ablation. Treatment with ablation can be approached percutaneously or laparoscopically. With the percutaneous approach, the lesion needs to be easily accessible for image-guided placement of the ablation probes. Proximity to major blood vessels and major bile ducts should be avoided given the heat sink effect in which an incomplete treatment occurs due to the cooling effect of major vessels. Dome lesions or those close to a main bile duct are also not ideal for ablation as the heat can cause thermal injury to the diaphragm or bile ducts. Ablation is not as effective for larger tumors due to the need for adequate margins, and, therefore, is only performed if a tumor is less than 3 cm in size with three or fewer separate tumors.

### 3.4. Locoregional Therapy for Downstaging and Bridge to Transplant

For patients undergoing LT, LRT with ablation, Yttrium-90 radioembolization (Y90), transarterial chemoembolization (TACE), or stereotactic body radiation therapy (SBRT) should be performed to treat the HCC lesion as a bridge to transplantation. The choice of LRT depends on the location, size, and number of HCC lesions. Ablative therapies, as previously mentioned, are commonly used for smaller lesions less than 3 cm in size and should not be located close to other organs, major vessels, and bile ducts. TACE provides a two-fold therapeutic approach: The first approach is arterial blockade which reduces or eliminates blood flow to the tumor, thus causing tumor ischemia and tumor necrosis. The second is the administration of a highly concentrated dose of chemotherapy to the lesion. Y90 involves the passage of a catheter through the hepatic artery, localized to the area of the tumor, where Y90 microspheres are released. These microspheres then slowly emit radiation into the tumor. All arterially directed therapies are relatively contraindicated in patients with bilirubin ≥ 3mg/dL due to the risk of hepatotoxicity and liver decompensation. LRT is relatively contraindicated in patients with higher CP B and CP C, with some case-by-case exceptions. SBRT has the advantage of treating small lesions, especially in “difficult-to-reach” locations, and can be used for “difficult-to-treat” lesions when TACE or Y90 have not been effective. Moreover, SBRT is able to accurately deliver a focused high-dose treatment to the targeted tumor, thus minimizing toxicity to normal surrounding organ structures. 

## 4. Non-Curative Approaches (Palliative/Tumor Control)

### 4.1. Locoregional Therapies

Patients with intermediate-stage BCLC B HCC who have multifocal disease and preserved liver function can receive LRT with arterially directed therapies (TACE/Y90) and/or SBRT (Table 2). Although TACE has historically been the primary treatment of choice for stage BCLC B/intermediate HCC, Y90 has become an accepted alternative therapeutic option. The decision on which intra-arterial-directed therapies to use will be dependent on center expertise and access. The goals of treatment are palliative, focusing on tumor control while maintaining quality of life and minimizing treatment-related toxicity.

### 4.2. TACE

TACE is recommended as first-line therapy for BCLC B/intermediate HCC in patients without vascular involvement. Improvements in OS have been clearly demonstrated in a meta-analysis of randomized controlled trials comparing TACE and best supportive care [45,46]. Liver tumors receive the majority of their blood supply from the hepatic artery. Both TACE and Y90 take advantage of the neovascularization of tumors and deliver chemotherapy or radiation treatment directly to the cancer by isolating the supplying hepatic artery using interventional radiology techniques.

TACE involves a two-step approach of intra-arterial injection of cytotoxic drugs to the cancer with subsequent embolization using an embolic agent that cuts off blood supply to the cancer, resulting in tumor necrosis. This conventional approach typically involves the use of doxorubicin or cisplatin emulsified in lipiodol (an oil-based radio-opaque contrast agent used as both a chemotherapeutic carrier and an embolic agent). By delivering chemotherapeutics directly to the tumor, this method delivers higher concentrations to the tumor without systemic toxicities. Subsequent new techniques involving the administration of drug-eluting beads (DEB) (embolic microsphere containing cytotoxic drugs) can be directed into the hepatic artery, allowing more sustained high concentrations directed to the tumor bed. Multiple randomized controlled trials comparing the OS, efficacy, and safety of conventional TACE with DEB-TACE have not shown any significant differences between the two techniques [47,48,49].

If the portal vein (PV) is compromised by a thrombus, then the liver becomes more dependent on the hepatic artery, and there can be a risk of hepatic infarction and liver failure. Therefore, Y90 is generally preferred when PV thrombus is present. Even with a patent PV, TACE can cause some hepatic injury and induce liver decompensation. For this reason, the NCCN has advised that bilirubin above 3mg/dL is a relative contraindication to TACE [50]. Due to immediate side effects related to the procedure, an overnight stay in the hospital is often necessary to monitor for post-embolism syndrome, which includes fever, pain, and nausea.

Treatment with TACE can be performed more than once. However, treatment that is deemed a TACE failure or refractory is when (1) the tumor lacks objective response post-treatment with > 50% viable disease after two TACE sessions; (2) new HCC has developed within the area of treatment zone after two TACE sessions; (3) AFP has not shown improvement despite two TACE sessions; and (4) there is progression of HCC with advancement of HCC staging, such as with vascular invasion or extrahepatic metastases [17]. Once patients are deemed as having TACE treatment failure, other alternative treatments should be considered, including systemic therapy.

### 4.3. Y90

Transarterial radioembolization is also performed by an interventional radiologist, but instead of chemotherapy, a radioactive isotope, Yttrium-90, is delivered intra-arterially. This procedure is typically performed in one session, but an initial mapping session is required to quantify the amount of hepatopulmonary shunting and gastroduodenal reflux. If excessive hepatopulmonary shunting or gastroduodenal reflux is present, there is a risk for radiation pneumonitis or gastric ulceration, and the procedure is contraindicated [51]. With the presence of PV thrombus, in contrast to TACE, Y90 has minimal embolic effect with a low risk of hepatic ischemia and therefore can be safely delivered [52]. Although the presence of PV thrombosis is not a contraindication to Y90, it is a negative prognostic marker and outcomes are worse for these patients [53]. Adequate liver function is required for this procedure, and pretreatment bilirubin values above 2mg/dL are a predictor of the risk for radiation-induced liver disease post-procedure [54]. Tolerability for Y90 is superior to that for TACE, especially with respect to abdominal pain, transaminitis, and time in the hospital [55]. 

The landmark LEGACY study demonstrated that treatment with Y90 is safe and effective for early HCC. This multi-center retrospective trial included 162 patients with CP A and a solitary HCC lesion less than 8 cm in size with a median lesion size of 2.7 cm. The study showed an ORR of 88.3% during a follow-up period of 29.9 months, with a 3-year OS of 86.6% [56]. Based on these results, the Food and Drug Administration approved the use of Y90 for HCC in 2021. In addition, in the prospective single-center RASER study, 29 patients with early HCC, who were not candidates for RFA, were treated with Y90 radiation segmentectomy with curative intent. ORR was 100% while CR was 90%. OS at 1-year and 2-years were 96% [57]. This study demonstrated that radiation segmentectomy was safe and effective for unresectable early-stage HCC with potential curative intent. (Table 3).

Limited studies have evaluated the safety and efficacy of Y90 and TACE [58,59]. The TRACE study is the largest prospective study to date comparing Y90 and DEB-TACE in a single-center randomized trial involving 72 patients with BCLC A or B, not eligible for surgery or ablation [58]. In patients receiving Y90, the median TTP was 17.1 months while patients receiving DEB-TACE had 9.5 m. For Y90, the median OS was 30.2 m and for DEB-TACE it was 15.6 m. The safety profile was similar between the two treatment arms. This study demonstrated that Y90 is associated with superior tumor control and better OS when compared with DEB-TACE (Table 3).

Lastly, treatment with Y90 using personalized dosimetry vs. standard dosimetry provides better radiologic response and improved survival with fewer adverse events. The concept of personalized dosimetry requires a delicate balance between adequate radiation dose to the tumor and preserving liver function. The DOSISPHERE study, a randomized multi-center phase II trial, evaluated 60 patients with BCLC B and C, achieving ORR in 71% and 36% for personalized and standardized dosimetry groups, respectively [60]. The median OS rates in the intention-to-treat analysis were 26.6 and 10.7 m for the personalized and standard dosimetry, respectively. Patients in the standard dosimetry group received 120 +/− 20 Gy to the perfused lobe. At least 205 Gy was targeted to the index lesion in the personalized dosimetry group, with less than 120 Gy to the non-tumor tissue (Table 3). The use of personalized dosimetry was further validated by the global TARGET study, a multi-center retrospective study of 207 patients with BCLC B/C treated with increased tumor-absorbed doses [61]. Patients receiving an increased tumor-absorbed dose were associated with improved ORR and OS.

### 4.4. External Beam Radiation

While Y90 delivers radiation internally to the tumor, external delivery of radiation is another treatment option for patients using SBRT. The overall survival (OS) and local tumor control rates are excellent, though most studies are either retrospective or observational. For small lesions, the local control and OS rates compare favorably to the rates seen with ablation [62]. As an advantage over ablation, SBRT can easily treat lesions regardless of proximity to the hepatic dome or blood vessels. However, the caudate lobe should be treated with caution, as edema or off-target effects can damage the neighboring bowel [63]. Hepatic toxicity is low, with rates reportedly less than 10%, and PV thrombosis is not a contraindication [64,65]. With increasing size of a lesion, the efficacy of SBRT decreases, and it is most effective in tumors less than 6 cm in diameter [65]. Emerging studies are also showing the potential benefit of SBRT in combination with TACE in the treatment of unresectable HCC with PV thrombosis. SBRT could achieve thrombus reduction or resolution, allowing PV flow restoration that will then allow TACE treatment [66,67].

### 4.5. Systemic Therapies

Patients with BCLC B or C HCC who have adequate performance status and liver function, but are no longer candidates for LRT, either due to disease burden or extrahepatic spread, should be considered for systemic therapy. Untreated BCLC B and BCLC C HCC portends a poor prognosis of approximately 9 months (m) and 3 m, respectively, and thus effective therapies are essential to improve outcomes [68]. HCC is resistant to conventional cytotoxic chemotherapy due to several complex molecular mechanisms, including autophagy activation, apoptosis evasion, expression of drug efflux pumps, enhancement of intracellular drug metabolism, and development of DNA repair mechanisms, among others [69]. The treatment options in 2023 for first-line treatment for HCC include targeted therapies such as multikinase inhibitors (MKI), anti-VEGF therapies, immune checkpoint inhibitors (ICIs), or combinations of these (Table 4).

## 5. Multikinase Inhibitors

Significant progress has been made in the advancement of systemic therapies in HCC treatment since the FDA approval of sorafenib in 2007. No effective systemic treatments were available until sorafenib, a tyrosine kinase inhibitor (TKI), was shown to be superior to placebo. HCC is a highly vascular tumor, and the signaling pathways promoting angiogenesis, such as VEGF, are critical in HCC tumor growth and metastatic potential [70]. The mechanism of action of the small molecule MKI against HCC is suppression of tumor growth, cell proliferation, differentiation, and angiogenesis through multiple complex pathways. Sorafenib inhibits VEGF, PDGFR, Raf, Ras, MEK, ERK, c-KIT, and RET, whereas lenvatinib inhibits VEGF, FGFRs, PDGFR, SCFR, KIT, and RET [71]. TKIs were the only option for over a decade due to many trials that failed to show superiority to sorafenib. Despite an improvement in OS with TKIs, the response rates were disappointing, and durability was lacking.

### 5.1. Sorafenib

In 2007, for the first time, systemic therapy was shown to improve outcomes over placebo in the first-line setting for advanced HCC. In the SHARP trial, investigators randomized 602 patients with advanced HCC to first-line therapy with either sorafenib 400 mg by mouth twice daily or to placebo [72]. The co-primary outcomes were OS and time to symptomatic progression, while the secondary outcomes were time to radiographic progression and safety measures. Patients could not be eligible for local therapies and had to have an ECOG performance status of 0-2 and CP A cirrhosis.

The OS primary outcome demonstrated a median survival of 10.7 m in the sorafenib group compared to 7.9 m with placebo. At one year, the survival rates were 44% and 33%, respectively, representing a 31% relative reduction in the risk of death (HR 0.69). The time to symptomatic progression primary outcome showed no difference in sorafenib and placebo, but a secondary endpoint of radiographic progression-free survival (PFS) was met with a longer radiographic PFS with sorafenib (5.5 vs. 2.8 m). Although the disease control rate was improved with sorafenib (43% vs. 32%), the objective response rates (ORR) were disappointing. Only 2% of patients achieved a partial response (PR) by RECIST. There were no complete responses (CR), and most patients had stable disease (SD) as their best response with sorafenib. Most patients enrolled in the SHARP trial were from Europe and Australia (88%) and about 10% were from North America. A second phase III study of sorafenib vs. placebo confirmed the efficacy of sorafenib in patients in the Asia-Pacific region, with a similar HR for death of 0.68 and similar poor ORR (PR in 3.3% vs. 1.3%) [73]. This trial also showed a slightly longer PFS (2.8 vs. 1.4 m) but no difference in time to symptomatic progression in the two groups.

Sorafenib was shown to have toxicities in 80% of patients compared to only 52% of placebo, but most side effects were mild with < 30% of patients having a grade 3–4 adverse event (AE). The most common AEs in the sorafenib group included diarrhea, fatigue, hand–foot syndrome (HFS), alopecia, and anorexia. Diarrhea and HFS were the most severe with 8% of patients having a grade 3 event for each. Sorafenib was considered well tolerated overall, with a permanent drug discontinuation rate due to AE of only 11% (compared to 5% in placebo).

### 5.2. Lenvatinib

In the ten years following the approval of sorafenib, many phase III trials with various drugs and combinations failed to show non-inferiority or superiority to sorafenib. Sorafenib remained the only option for first-line treatment until 2018 with the addition of lenvatinib, a potent MKI, to the treatment arsenal. The REFLECT study was an open-label, phase 3, non-inferiority trial that compared lenvatinib to sorafenib in first-line unresectable HCC [74]. Nine hundred fifty-four patients with CP A cirrhosis and ECOG 0–1 were included. Dosing was based on body weight, with patients at least 60 kg receiving 12 mg daily and patients less than 60 kg receiving 8 mg daily. Patients were excluded from the study if they had main PV invasion, 50% or more liver involvement, or uncontrolled hypertension. The enrollment took place in 20 countries; approximately two-thirds of patients were from the Asia-Pacific region and one-third were from the Western region. The primary endpoint was OS, and this was first tested for non-inferiority and then for superiority. Secondary endpoints included PFS, time to progression (TTP), ORR, and quality of life (QOL) measurements.

The primary endpoint for OS was met for non-inferiority, but not for superiority. The median OS was 13.6 and 12.3 m in the lenvatinib and sorafenib groups, respectively, and this difference was not statistically significant. Lenvatinib demonstrated superiority in all secondary endpoints, including PFS, TTP, and ORR. Nearly a quarter (24.1%) of patients in the lenvatinib arm showed an objective response (mRECIST, investigator review) with 23% PR and 1% CR. In the sorafenib arm, the response rate was lower, with only 9.2% having an objective response (mRECIST, investigator review), 9% with a PR, and less than 1% with a CR. The disease control rate was higher in the lenvatinib arm, and more patients had progressive disease as the best response in the sorafenib arm.

The rate of AEs was similar in the two groups, though the side effect profile was different. The most common AE for sorafenib was palmar–plantar erythrodysaesthesia (PPE) (any grade 52%, 11% grade 3–4). The sorafenib arm also saw more alopecia (25% vs. 3%) and diarrhea (46% vs. 39%). The patients on lenvatinib had more significant hypertension, proteinuria, and hypothyroidism. Fatigue, anorexia, and diarrhea were common in both treatment arms. Less than 10% of patients had to completely stop therapy due to an AE, but patients in the lenvatinib arm did have a slightly longer time on treatment (5.7 vs. 3.7 m). In this trial, just slightly more than a third of patients went on to receive second-line therapy, underscoring the importance of choosing the optimal first-line treatment for patients.

Additional TKIs have been approved by the FDA and are indicated as second-line treatments. Regorafenib, cabozantinib, and ramucirumab have been tested in the second-line setting and showed superiority over the placebo. Ramucirumab is given to biomarker-selected populations; it is only approved in patients whose AFP is at least 400 ng/mL. If an ICI is used in the first-line setting, sorafenib or lenvatinib can be considered as options for second-line treatment.

## 6. Immunotherapy

ICIs have demonstrated durability in numerous solid tumors, and the immunobiology of HCC lends itself to therapeutic intervention targeting the immune cells. The presence of tumor-infiltrating lymphocytes in HCC tumors correlates with outcome, suggesting that the immune responses could be important in treating HCC [75]. Immune checkpoint proteins are involved in the control of a person’s immune response, keeping the immune system in check. There are a number of these proteins on T cells, including PD1, CTLA4, TIGIT, and LAG3, each of which can be inhibited by ICIs. When ICIs inhibit these checkpoints, it allows the patient’s immune response to activate and destroy cancer cells. The challenges with ICIs in HCC include the fact that cancer cells typically begin in an environment of chronic inflammation, and many immune cells in the liver are involved in maintaining and promoting tolerance to neo-antigens. The tumor immune microenvironment of HCC can dampen the host immune response and promote tolerance [76].

Although single-agent checkpoint inhibition with PD1/PDL1 was not superior to sorafenib, the response rate, durability, and safety signals were encouraging [77,78]. In the Checkmate 459 study, nivolumab was compared to sorafenib in the first-line setting, and despite no difference in survival, the response rate by RECIST was higher (15% vs. 7%) and the rates of grade 3–4 AE were lower (22% vs. 49%) [77]. Since that time, the use of ICIs in combination with other agents to harness the immune response has proven to be more successful. These combinations have redefined the treatment options and have largely supplanted TKIs as the preferred first-line option for most patients.

### 6.1. Atezolizumab and Bevacizumab

Bevacizumab combined with atezolizumab was FDA-approved in May 2020, and it was the first systemic treatment option found to improve survival over sorafenib in over a decade. Bevacizumab is a VEGF monoclonal antibody, and when used in combination with ICIs, it can change the microenvironment to an immune stimulatory environment by improving priming and activation of T cells, tumor infiltration of T cells, and inhibiting cells that lead to immune suppression [79]. The mechanisms of anti-VEGF antibodies combined with PD1/PDL1 antibodies lead to a synergistic effect to achieve better outcomes than with ICIs alone [80].

The IMbrave 150 trial was an open-label, phase 3 trial that randomized 501 patients with unresectable HCC to atezolizumab plus bevacizumab or sorafenib [81]. The co-primary endpoints were OS and PFS. Secondary endpoints included ORR, duration of response, and time to deterioration of QOL. This was a global study with 40% of patients enrolling from Asia and Japan. Patients with CP A cirrhosis and ECOG 0–1 were included. Due to the potential bleeding risk with bevacizumab, an updated EGD within 6 m of treatment was required, with treatment of varices as per standard of care. Patients with untreated or incompletely treated varices with a high risk of bleeding were excluded. A quarter of patients had varices, and some had untreated varices at baseline (11–14%). Patients were also excluded if they had uncontrolled hypertension, recent hemoptysis, or were on full-dose anticoagulation. In contrast to prior studies, high-risk patients with main PV invasion or involvement of at least 50% of the liver were included.

The primary outcomes were met with an improvement in OS at 12 m (67.2% vs. 54.6%) and an improvement in PFS by 2.5 m (6.8 vs. 4.3 m). With 12 m of additional follow-up, the median OS for atezolizumab and bevacizumab was the longest median OS for any systemic therapy at the time of the publication (19.2 vs. 13.4 m) [82]. The response rates by mRECIST were significantly improved with atezolizumab and bevacizumab (33.2% vs. 13.3%, *p* < 0.001), and there was an impressive 10.2% complete response rate in the combination arm. The disease control rate was improved (72% vs. 55%), and patients were able to stay on treatment longer. As is typical with ICIs in other cancers, the durability of response was improved in the combination arm (duration of response not reached vs. 6.3 m) [83].

Atezolizumab and bevacizumab were well tolerated, and the mean duration of treatment was more than double the time on treatment with sorafenib (7.4 vs. 2.8 m). Most patients did experience an AE, but few patients had to discontinue treatment due to side effects in either arm. Diarrhea and PPE were significantly more common in the sorafenib arm. Proteinuria and hepatitis were more common in the combination arm [82]. This regimen has supplanted TKIs as the standard of care for eligible patients given the improved survival, response rates, durability, and safety profile.

### 6.2. Tremelimumab and Durvalumab

The cytotoxic T-lymphocyte-associated antigen 4 (CTLA-4) immune checkpoint is distinct from the PD1/PDL1 checkpoints, and blocking both leads to complementary effects on antitumor immune responses [84]. Data from a phase 1b study showed maximum expansion of T cells after a single dose of CTLA-4 antibody treatment that did not increase further after additional doses [85]. In addition, toxicity from CTLA-4 antibodies often comes after repeated exposure. Therefore, a single dose of tremelimumab was tested in combination with durvalumab (PD-L1 inhibitor). This dual ICI therapy invoked responses that were not seen with single-agent ICIs [86].

The HIMALAYA trial randomized 1171 patients with unresectable HCC requiring first-line systemic therapy to the combination of tremelimumab and durvalumab, durvalumab monotherapy, or sorafenib [87]. The STRIDE (single tremelimumab, regular interval durvalumab) regimen consisted of a single 300 mg dose of tremelimumab in addition to durvalumab 1500 mg every 4 weeks. The primary objective was to evaluate OS for STRIDE vs. sorafenib, and the secondary endpoint was to evaluate the noninferiority of durvalumab vs. sorafenib. A second combination regimen, T75 + D, consisted of tremelimumab 75 mg every 4 weeks for four doses plus 1500 mg of durvalumab every 4 weeks, but enrollment was closed to this arm when a phase 2 study demonstrated no difference in efficacy compared to durvalumab monotherapy [19]. Patients in this study had BCLC B or C HCC, were CP A, and were ineligible for LRT. Patients were excluded if they had a thrombosis in the main PV, prior LT, or a history of an autoimmune disease.

The primary endpoint of OS in the HIMALAYA trial was met. Patients receiving the STRIDE regimen had a median OS of 16.4 m, compared to 13.8 m with sorafenib. With longer follow-up, a quarter of patients were still alive at 4 years (4-year OS 25.2% vs. 15.1%) [88]. The secondary endpoint for noninferiority of durvalumab compared to sorafenib was also met, but the superiority of durvalumab was not significant. The median TTP in each group was similar (5.4 vs. 5.6 m). The discrepancy in PFS and OS results may be due to disease stabilization after initial progression in patients getting ICIs. The investigators allowed patients to be rechallenged with tremelimumab beyond radiographic progression if they met certain criteria, including investigator-assessed benefit to treatment, no threat to vital organs, and progression that did not occur after a PR or CR. The disease control rate was similar across the three arms, but the ORR by RECIST was higher in the immunotherapy arms (20.1%, 17.0% vs. 5.1%). Twelve patients (3.1%) in the STRIDE arm had a CR, compared to none in the sorafenib arm.

Treatment with the STRIDE regimen was well tolerated. Treatment-related grade 3 or 4 AEs were seen most often in the sorafenib group (36.9%), leading to a dose delay or discontinuation in nearly half of patients. In contrast, grade 3 or 4 treatment-related AEs were seen less frequently in the STRIDE regimen (25.8%), with less than a third of patients requiring dose delay or discontinuation. As expected, grade 3 or 4 immune-mediated AEs were more common in the STRIDE regimen compared to durvalumab (12.6% vs. 6.2%), and the requirement of high-dose steroids was also more common in the STRIDE regimen compared to durvalumab alone (20.1% vs. 9.5%).

### 6.3. Combination of ICIs with Multikinase Inhibitors

Using VEGF TKIs to modulate the immunosuppressive microenvironment and increase the efficacy of ICIs is another strategy that has been tested as a first-line treatment of HCC; however, the studies evaluating these combinations have shown mixed results. In LEAP 002, pembrolizumab and lenvatinib were compared to lenvatinib, but the primary endpoints of OS and PFS were not met. Although the median OS for the combination was an impressive 21.2 m, the control arm did remarkably well (19 m), so the difference was not statistically significant [89]. Similarly, in the COSMIC-312 study, cabozantinib plus atezolizumab improved PFS (HR 0.63) compared to sorafenib, but it did not show any difference in OS (15.4 vs. 15.5 m) [90]. Again, the control arm did remarkably well with an almost 50% increase in survival for sorafenib compared to what was seen in the original SHARP trial. More patients in the sorafenib arm received subsequent therapy (37% vs. 20%), with a higher percentage of patients receiving immune therapy in the second-line setting, which could explain the longer-than-expected OS in the sorafenib arm.

In 2023, the CARES-310 study showed a significant improvement with the combination of camrelizumab (a PD1 inhibitor) and rivoceranib (an oral VEGFR TKI) compared to sorafenib [91]. This phase 3 trial randomized 543 patients with primary outcomes of OS and PFS. Both primary outcomes were met with a PFS of 5.6 vs. 3.7 m (HR 0.52) and a median OS of 22.1 vs. 15.2 m (HR 0.62). Although the PFS was only improved by less than 2 m, the 7 m difference in OS was notable with the longest median OS published to date.

This trial was an international trial, but most patients enrolled were from Asia, with only 17% of patients from non-Asian countries. This contrasts with HIMALAYA and IMbrave150 studies in which most patients were from non-Asian countries. Given this global distribution, a large proportion of patients in CARES-310 had viral hepatitis as the cause of their cirrhosis. Only 15% of patients had a non-viral cause of cirrhosis in the CARES-310 study, whereas IMbrave 150 and HIMALAYA had 30–40% of patients with non-viral causes. The combination of camrelizumab and rivoceranib was difficult to tolerate, and 81% of patients had a grade 3–4 AE (compared to 52% with sorafenib). The OS was remarkable, but given the high incidence of AE, and the relatively low numbers of Western patients with non-viral causes of cirrhosis, it remains to be seen if these results will be generalizable or change practice in the West. At the time of this writing, this combination is not FDA-approved, but the FDA has accepted a new drug application for camrelizumab and rivoceranib for first-line treatment in patients with metastatic HCC.

### 6.4. Future Directions in the Era of Immunotherapy

As HCC therapeutics have entered into the new era of immunotherapy, there is great interest in the safety and efficacy of immunotherapy in early- and intermediate-stage HCC as well as the role of ICIs in combination with LRT.

Given the high rates of recurrence with resection or ablation, adjuvant therapy has been an area of investigation. Sorafenib did not improve recurrence-free survival (RFS) rates when administered post-operatively [92]. A recent trial, which has not yet been published as of this writing, was shown to improve outcomes in the adjuvant setting and has the potential to change practice. The IMbrave050 trial was an open-label phase III randomized clinical trial that compared atezolizumab and bevacizumab to active surveillance in patients with HCC at high risk of recurrence after ablation or resection [93]. Patients were treated with atezolizumab and bevacizumab every 3 weeks for 17 cycles, or 1 year. High-risk features included size over 5 cm, more than three tumors, microvascular or minor macrovascular invasion, or grade 3/4 pathology. The primary endpoint of RFS was met at the interim analysis with an HR of 0.72 (*p* = 0.012) and 12 m RFS of 78% and 65%, respectively. Longer follow-up is needed to determine if the RFS benefit will be maintained in subsequent analyses or if progression was merely delayed with one year of adjuvant therapy. Various phase 3 trials are ongoing to evaluate ICIs in early-stage HCC after resection or ablation.

For intermediate-stage HCC, ongoing trials are also evaluating combination therapy with TACE or SIRT with ICIs in combination with synchronous or on-demand intra-arterial therapies. In addition, future considerations in LT include the potential use of ICIs for the purpose of downstaging or as a bridge to transplantation, thus allowing eligibility for HCC MELD exception points. The timing of discontinuing ICIs prior to transplantation remains unclear. Neoadjuvant studies are also ongoing. While early findings are promising for the role of ICIs prior to transplantation, larger trials are needed to ensure safety and efficacy prior to implementing this high-risk strategy in routine clinical practice.

Drugs with novel mechanisms are also of interest. The recent success of tiragolumab, an anti-TIGIT antibody, in addition to atezolizumab and bevacizumab in the phase Ib/II MORPHEUS-liver study, has paved the way for the ongoing phase 3 trial IMbrave152 looking at this triplet combination. New combinations of previously evaluated drugs, such as adding ipilimumab to atezolizumab and bevacizumab, are also being tested. Trials looking at completely novel therapeutics (vaccines and CAR-T) are also ongoing.

The future role of combination therapies with ICIs in the treatment of CP B cirrhosis patients remains unclear, given the concerns over safety in this patient population. Studies with nivolumab thus far have shown it to be safe and effective. Recent real-world data provide preliminary evidence for the safety and efficacy of atezolizumab plus bevacizumab in patients with CP B cirrhosis [94]. See Table 5 for a selected list of ongoing clinical trials.

As clinical trials continue to focus on ICIs and other novel agents, challenges remain in the understanding of the molecular heterogeneity of HCC and the liver tumor microenvironment. Biomarkers are needed to stratify patients and predict how they will respond to certain therapies. Additional therapeutic approaches may be needed to increase tumor susceptibility to ICIs in patients who are less likely to respond, whether it be due to the etiology of cirrhosis or to other reasons for a dampened immune response. The treatment paradigm for HCC is evolving, and the future model is envisioned to be one of precision medicine and personalized care. This would involve the ability to identify biomarkers for early detection, treatment response, and disease surveillance with the incorporation of clinical, radiologic, and biochemical data in the era of machine learning and artificial intelligence.

## 7. Best Supportive Care: Incorporation of Non-Hospice Palliative Care in HCC

Patients with HCC commonly have preexisting cirrhosis, and this dual diagnosis increases the complexity of their care. This patient group bears considerable physical, psychosocial, and financial burdens, with caregiver burnout as well as possible stigmatization. Symptoms from liver decompensation that can occur with HCC include ascites, variceal hemorrhage, hepatic encephalopathy, sarcopenia, and frailty. These patients are also faced with symptoms related to their tumor, extra-hepatic spread, or effects of treatment. The most common symptoms faced by patients with HCC are abdominal pain, fatigue, anorexia, nausea/vomiting, and ascites [95]. Liver cancer also ranks in the top three cancers for high prevalence of depression and anxiety [96]. As the disease progresses and the symptom burden increases, the role of PC becomes more evident (Figure 3).

There is a growing recognition in the hepatology community regarding the importance of early intervention of palliative care (PC) for HCC patients, whether the goals of therapy are curative or non-curative. Early intervention with PC even in earlier stage disease can assist with symptom management, advanced care planning, and psychosocial support. PC has been historically underutilized in patients with HCC, and over a quarter of patients with advanced-stage HCC never enter hospice care before the time of their death [97]. Various barriers to referral to PC include prognostic uncertainty, the unpredictable clinical trajectory of cirrhosis, lack of time for these discussions, stigma and biases from the patient or caregiver, and the misconception that PC is associated with “giving up” [98]. A new framework for HCC care involving partnerships with PC, hepatology, and the transplant team is greatly needed, and ensuring that referral to PC is not synonymous with stopping active therapies or disease-directed therapies.

Creating a new clinical model of practice between the hepatology/transplant team and PC will require a paradigm shift in clinical practice that includes incorporating PC providers in the multidisciplinary team model. This new care model is designed to provide patient-centered supportive care, whether treatment goals are curative or non-curative. The incorporation of PC in HCC management can provide benefits to overall care, including improving patient quality of care and QOL as well as supporting caregiver and care teams. In addition, the PC team can initiate early discussions of advance care planning but would approach it differently in patients who are pursuing curative-intent therapy, including LT, in contrast to patients with non-curative palliative goals of care or those who require hospice care [99]. Models of care involving the integration of PC are part of routine practice in several end-stage diseases, including advanced cancer, chronic kidney failure, and congestive heart failure. This model of care has been shown to increase survival, decrease hospitalizations, and improve patient QOL [100]. There are currently limited evidence-based data to provide recommendations regarding PC involvement specifically in HCC care [101]. Further research is needed to better understand the timing of PC referral, intervention, and outcomes of HCC patients receiving PC.

## 8. Conclusions

HCC is a particularly lethal malignancy with a prevalence that varies according to the global region and risk factors. The highest global burden of HCC is in the East, with HBV as the most common cause in Asia as well as sub-Saharan Africa. In the West, non-viral causes such as MASLD and ALD are more common, and metabolic causes are increasing worldwide in parallel with the obesity epidemic.

The complexity of HCC care includes the management of not only the cancer but the underlying liver disease, and, therefore, it should be managed by a multidisciplinary team ideally in a co-located clinic at a liver transplant center. For patients with early-stage disease, curative approaches are possible through surgical resection, ablation, or LT. If portal hypertension or tumor distribution precludes surgery, then LT should be considered as it provides the best outcomes of all of the curative therapies by addressing both the cancer and the underlying liver disease.

For intermediate-stage patients, various LRTs can be considered, including ablation, SBRT, Y90, and TACE. For advanced disease, systemic therapies are offered when there is recurrence after LRT, or if there is extensive or extrahepatic disease. For over a decade, TKIs were the only option, but now ICI combinations are the preferred option unless contraindications preclude initiation. First-line systemic therapies include combination anti-VEGF antibody or CTLA inhibitor and PD1/PDL1 inhibitors, monotherapy with single-agent immunotherapy, or TKI monotherapy. Comorbidities, patient factors, performance status, liver transplant status, need for response, and patient preferences need to be considered when choosing a first-line treatment regimen.

Lastly, an emerging paradigm shift in HCC care involves the early incorporation of the PC team and the importance of its role in the multidisciplinary team care model. Early incorporation of PC services (non-hospice) in management should be considered. Whether the goals of therapy are curative or non-curative, this model of care can provide patient-centered supportive care with benefits that include improving the quality of patient care and patient QOL as well as supporting caregivers and care teams. More studies are needed to further evaluate and better understand the role of PC care in HCC and how this may affect patient outcomes and healthcare utilization.

The treatment landscape for HCC continues to evolve. Future directions in the treatment of HCC include incorporating systemic therapies into earlier stages of the disease, novel therapeutic options, as well as novel combinations involving both ICIs and LRTs. As we strive to further understand the molecular heterogeneity of HCC and its liver tumor microenvironment with its associated tumor biomarkers, we pave the way for the future of precision medicine and personalized care, propelled by advancements in artificial intelligence.

## Figures and Tables

**Figure 1 cancers-16-00666-f001:**
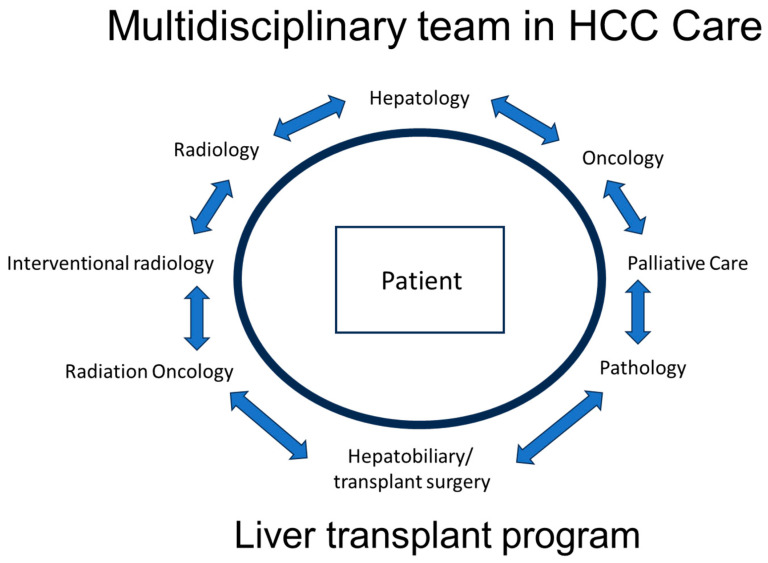
Multidisciplinary team model in HCC care.

**Figure 2 cancers-16-00666-f002:**
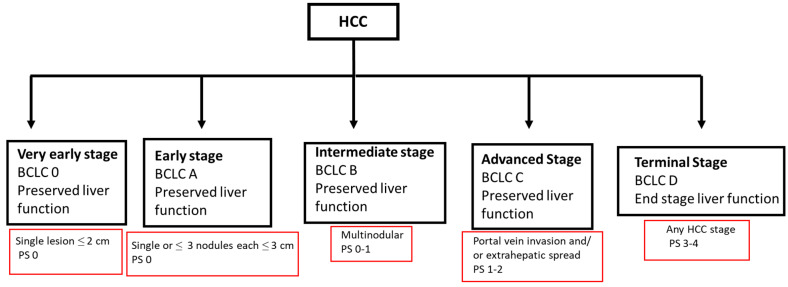
BCLC HCC staging 2022. BCLC, Barcelona clinic liver cancer staging system; HCC, hepatocellular carcinoma; PS, performance status score.

**Figure 3 cancers-16-00666-f003:**
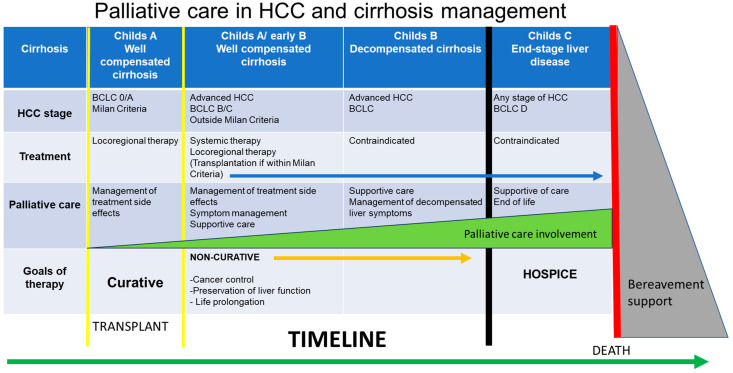
Incorporation of palliative care in HCC care involves consideration of the cirrhosis stage, HCC stage, treatment, and goals of therapy. The level of involvement of palliative care will vary based on these factors.

**Table 1 cancers-16-00666-t001:** Curative-intent treatments.

	Recurrence Rate	Overall Survival	Ideal Candidate	Exclusion	Key Issues
Ablation	73–80% [22,23]	70% [22]	– very small HCC ≤ 2 cm in size – not a surgical candidate – location easily accessible via the percutaneous route	– adjacent to major blood vessels or bile ducts due to heat sink effect – typically not used over 3 cm in size – dome lesions close to the diaphragm	– if a patient is a transplant candidate with a very small HCC, observation until >2 cm may be recommended in order to obtain MELD exception points
Resection	70% [24]	70–80% [24]	– no cirrhosis or CP A cirrhosis without clinically significant portal hypertension – solitary mass – location will allow for an adequate liver remnant after resection	– clinically significant portal hypertension – multifocal/bilobar disease	– if the size of the future liver remnant is a concern, preoperative portal vein embolization can be performed to induce hypertrophy of the future liver remnant
Transplant	10–15% [25,26]	80%	– cirrhosis severity precludes resection – within the Milan criteria	– not expected to survive a major surgery	– expanded criteria available if the patient is not within the Milan criteria, with regional variations – downstaging to Milan is possible with local regional therapies

**Table 2 cancers-16-00666-t002:** Liver-directed therapies.

	Advantages	Disadvantages
Ablation	– curative – well tolerated	– limited to small lesions, ideal for <3 cm – must be mindful of location (avoid dome lesions, adjacent to major vessels or bile ducts)
Y90	– can be used in the presence of portal vein thrombosis – outpatient procedure performed in two sessions (one mapping session and one treatment session) – well tolerated	– can be expensive – must pass the mapping procedure requirements (to avoid hepatopulmonary shunting or reflux)
TACE	– one-time treatment – recommended as first-line liver-directed therapy in the treatment algorithm for BCLC stage B patients	– overnight stay in the hospital is required to monitor for post-procedure pain and complications. – cannot be used in patients with portal vein thrombosis
External beam radiation	– minimal risk of liver damage – can be used in the presence of portal vein thrombosis – dome lesions can be treated	– often multiple days of treatment – caution advised if the bowel is in close proximity (caudate lobe lesions)

**Table 3 cancers-16-00666-t003:** Selected landmark studies for Y90.

Study	Design	N	Clinical Criteria	Radiologic Response	Survival	Adverse Events
LEGACY 2021	Multicenter, retrospective, noncomparative	162	-CP A cirrhosis -Solitary HCC lesion up to 8 cm (median size 2.7 cm)	TARE radiation segmentectomy Objective response rate: 88.3% mTTP: not reached	mOS: 57.9 mo 2-y OS: 94.8% 3-y OS: 84.6%	19.1%
TRACE 2022	Single- center, randomized controlled trial	72	BCLC B	TARE vs. DEB-TACE mTTP (ITT): 17.1 vs. 9.5 m mPFS:11.8 vs. 9.1 m	TARE vs. DEB-TACE mOS (ITT): 30.2 vs. 15.6 m	TARE vs. DEB-TACE: 39% vs. 53%
RASER 2022	Prospective, single, center, noncomparative	29	Very early/early HCC Not candidate for RFA Curative intent	TARE Objective response: 100% Complete response: 90%	1-y OS: 96% 2-y OS: 96%	7%
DOSISPHERE 2022	Randomized, multicenter phase II trial	60	BCLC B/C Non-resectable	Personalized dosimetry vs. standard dosimetry: mPFS (ITT): 6.0 vs. 3.4 m 3-mo ORR (ITT): 71 vs. 36%.	Personalized dosimetry vs. standard dosimetry: mOS (ITT): 26.6 vs. 10.7 mo. 1-y OS: 65.5% vs. 44.8% 2-y OS: 53.3% vs. 22.3%	Personalized dosimetry vs. standard dosimetry: 20% vs. 33%

mTTP: median time to progression; mPFS: median progression free survival; ORR: overall response rates; OS: overall survival; ITT: intention to treat analysis.

**Table 4 cancers-16-00666-t004:** Selected landmark studies for systemic therapy.

Study	Design	N	Intervention vs. Control	ORR Intervention vs. Control	DCR Intervention vs. Control	Survival (Months)
SHARP	Randomized, double-blind, placebo-controlled, phase III	602	Sorafenib vs. placebo	RECIST: 2% vs. 1%	RECIST 43% vs. 32%	10.7 vs. 7.9
REFLECT	Randomized, open-label, non-inferiority phase III	954	Lenvatinib vs. sorafenib	RECIST: 18.8% vs. 6.5% mRECIST (investigator review) 24.1% vs. 9.2% mRECIST (masked independent imaging review) 40.6 vs. 12.4%	RECIST: 72.8% vs. 59.0% mRECIST (investigator review) 75.5% vs. 60.5% mRECIST (masked independent imaging review) 73.8% vs. 58.4%	13.6 vs. 12.3
IMbrave 150	Randomized, open-label, phase III	501	Atezolizumab/ bevacizumab vs. sorafenib	RECIST: 27.3% vs. 11.9% mRECIST: 33.2% vs. 13.3%	RECIST: 73.6% vs. 55.3% mRECIST: 72.3% vs. 55.1%	19.2 vs. 13.4
HIMALAYA	Randomized, open-label, sponsor-blind, phase III	1171	Durvalumab/ tremelimumab vs. durvalumab vs. sorfenib	RECIST: durva/treme 20.1% vs. durva 17.0% vs. sorafenib 5.1%	RECIST: 60.1% vs. 54.8% vs. 60.7%	16.43 vs. 16.56 vs. 13.77

ORR: objective response rate; DCR: disease control rate; durva: durvalumab; treme: tremelimumab.

**Table 5 cancers-16-00666-t005:** Selected ongoing clinical trials.

Intervention	Study Population	Completion Date	Design	Clinical Trials ID
TACE with Tislelizumab as adjuvant therapy	Resectable HCC	December 2024	Phase 2	NCT04981665
Lenvatinib and TACE and camrelizumab vs. lenvatinib alone	BCLC C patients with the goal of conversion resection	1 December 2025	Phase 3	NCT05738616
Neoadjuvant and adjuvant lenvatinib	HCC patients receiving curative-intent percutaneous ablation with high-risk features for recurrence	4 May 2025	Phase 2	NCT05113186
Neoadjuvant Tislelizumab +/− lenvatinib	Resectable HCC	1 December 2025	Phase 2	NCT04615143
SIRT with tremelimumab and durvalumab	Resectable HCC	1 October 2025	Phase 1	NCT05701488
Dendritic cell vaccine and nivolumab	Resectable HCC	May 2025	Phase 2	NCT04912765
T cell therapy	Resectable HCC	30 June 2024	Phase 1	NCT05352646
Anti-PD-1 inhibitor (tislelizumab, pembrolizumab, or nivolumab) and local therapy	HCC beyond Milan criteria, undergoing downstaging for transplant	1 August 2028	Phase 2	NCT05475613
Atezolizumab, bevacizumab +/− tiragolumab	Locally advanced or metastatic	1 September 2026	Phase 3	NCT05904886
T cell therapy	Advanced HCC expressing GPC3	31 December 2025	Phase 1	NCT05003895

## Data Availability

The data presented in this study are available in this article.

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
