# Peer review of "Management of Hepatocellular Carcinoma in 2024: The Multidisciplinary Paradigm in an Evolving Treatment Landscape"

_cancers, 2024, doi:10.3390/cancers16030666_

Round 1

Reviewer 1 Report

Comments and Suggestions for Authors

Interesting review on the importance of multidisciplinary approach for HCC patients.

Some tables highligthing the most important studies in this field would be useful.

The authors should comment also on the important of this approach for HCC prevention, for example in MAFLD-related HCC (in this regard mention the important of bariatric approach, citing the recent SRMA PMID: 33721336 )

Following the previous point, mention also the approaches for HCC screening in a multidisciplinary setting

The authors should mention that ablative treatments can be also surgical, not only percutaneous

Author Response

We thank the reviewer for their comments. Please see below our responses in bold black. We have attached the updated manuscript and tables 3 and 4 (at end of manuscript)

Review 1:

Comments and Suggestions for Authors

Interesting review on the importance of multidisciplinary approach for HCC patients.

Some tables highligthing the most important studies in this field would be useful.

We thank the reviewer for their feedback.

We have added a table with relevant systemic therapy trials. (Table 4). 

We have added a table of selected landmark trials for Y90 (table 3). 

With the addition of Table 3 , we have added additional texts to better clarify between TACE (Conventional and DEB-TACE)  and Y90, (line 262-265/ pg 7, line 269-272/ 277-288, 290-291/ pg 8,  line 307-308/ page 9).

We have also added additional text regarding these selected landmark trials and their relevance in logoregional therapy that are mentioned in Table 4. (line 314- 348/ pg 9)

The authors should comment also on the important of this approach for HCC prevention, for example in MAFLD-related HCC (in this regard mention the important of bariatric approach, citing the recent SRMA PMID: 33721336 )   

Given the focus of this review paper is on management approaches and strategies, we believe HCC prevention is beyond the scope of this review and should not be added.

Following the previous point, mention also the approaches for HCC screening in a multidisciplinary setting.  This review paper is focused on the latest and future treatments/ management for HCC. The topic of screening for HCC is beyond the scope of this paper. However, we have added a small section on surveillance. (Line 114- 139/ pg 4)

The authors should mention that ablative treatments can be also surgical, not only percutaneous.

We appreciate the reviewer’s input on mentioning this approach, and agree that surgical management can certainly by used as well. This comment has been added. (Line 229-230/ Pg 7)

Reviewer 2 Report

Comments and Suggestions for Authors

The review by Kinsey and Lee deals with current perspectives on treatment strategies for HCC.

As a general comment, while this review has a very broad scope, the intrinsic risk is an overly simplistic description of some contents, making it less informative. Furthermore, several statements should be supported by specific references, in particular when divergent approaches are made possibile for the same condition. For instance, this is well depicted by the surgical approach according to EASL/AASL guidelines and Eastern guidelines.

Futhermore, most of liver–directed therapies are described without any reference to their clinical indications.

Specific comments:

1) Curative resection (page 3 and 4). It is stated that resection is not recommended in cases with vascular invasion, however the authors fail to acknowledge the Eastern guidelines regarding the possibility of liver resection in selected patients with MVI. Moreover, resective surgery in patients with multinodular disease is not recommended by EASL/AASL guidelines. Nevertheless, for patients with 2–3 nodules, both the Eastern and Italian multisociety guidelines (https://doi.org/10.1016/j.dld.2023.10.029) and centers expert in liver surgery do not preclude a surgical option which potentially outperforms TACE.

2) Page 7: Y90, SBRT, TACE: provide their clinical indications.

3) Provide radiological assessment criteria (RECIST vs mRECIST) in Sharp and Reflect trials.

4) Table 3 is not informative. I would delete the last three rows as they are confusing.

5) Table 4 is not mentioned in the text. I guess it is meant to show ongoing trials with ICI for intermediate stage HCC, but the authors fail to mention current results with ICI-based studies in resectable HCC (Kaseb AO, Hasanov E, Cao HST, et al. Perioperative nivolumab monotherapy versus nivolumab plus ipilimumab in resectable hepatocellular carcinoma: a randomised, open-label, phase 2 trial. Lancet Gastroenterol Hepatol 2022; published online Jan 19. https://doi.org/S2468-1253(21)00427-1).

Author Response

We thank the reviewer for their comments. We have attached the updated manuscript with tables (3-4) at end of manuscript.

Please see below our responses in bold black. 

Review 2:

The review by Kinsey and Lee deals with current perspectives on treatment strategies for HCC.

As a general comment, while this review has a very broad scope, the intrinsic risk is an overly simplistic description of some contents, making it less informative. Furthermore, several statements should be supported by specific references, in particular when divergent approaches are made possibile for the same condition. For instance, this is well depicted by the surgical approach according to EASL/AASL guidelines and Eastern guidelines.

Futhermore, most of liver–directed therapies are described without any reference to their clinical indications.

Specific comments:

1)Curative resection (page 3 and 4). It is stated that resection is not recommended in cases with vascular invasion, however the authors fail to acknowledge the Eastern guidelines regarding the possibility of liver resection in selected patients with MVI. Moreover, resective surgery in patients with multinodular disease is not recommended by EASL/AASL guidelines. Nevertheless, for patients with 2–3 nodules, both the Eastern and Italian multisociety guidelines (https://doi.org/10.1016/j.dld.2023.10.029) and centers expert in liver surgery do not preclude a surgical option which potentially outperforms TACE.

We appreciate the reviewer’s input regarding other guidelines that may approach surgical resection differently than AASLD and EASL. We agree that in certain  cases treatment approach may require going outside of guidelines particularly when patients are not transplant candidates or wish not to pursue transplantation.

Our liver cancer/ transplant program also pursues resection beyond standard guidelines when indicated and on a case by case basis. Although the topic of resection and its controversies are not the large focus of this paper, we have added a section regarding the approaches that the Eastern and Italian Multisocieties guidelines would recommend. (Lines 167-185/ Pg 5,6)

2)Page 7: Y90, SBRT, TACE: provide their clinical indications.

Yes, the clinical indications are listed under the locoregional therapies headings as stated line 260, 261/ pg 7.

Provide radiological assessment criteria (RECIST vs mRECIST) in Sharp and Reflect trials.

We appreciate the authors input regarding this. We placed these criteria in the text when applicable (line 404, 438, 440/ pg 11, line 475/ pg 12, line 505, 549/ Pg 13

4) Table 3 is not informative. I would delete the last three rows as they are confusing.

We have deleted this table.

5) Table 4 is not mentioned in the text. I guess it is meant to show ongoing trials with ICI for intermediate stage HCC, but the authors fail to mention current results with ICI-based studies in resectable HCC (Kaseb AO, Hasanov E, Cao HST, et al. Perioperative nivolumab monotherapy versus nivolumab plus ipilimumab in resectable hepatocellular carcinoma: a randomised, open-label, phase 2 trial. Lancet Gastroenterol Hepatol 2022; published online Jan 19. https://doi.org/S2468-1253(21)00427-1).

We appreciate the reviewers input regarding this. Table 4 was suppose to be referred to under the section of “Future directions” but was placed in the wrong section in the review.

Table 4 (now named table 5) is now mentioned in the text in the subheading “future directions.” It is a table looking at upcoming trials.

We did not specifically discuss neoadjuvant therapies in detail in the review paper, so did not mention the results of this phase 2 NA trial.

Reviewer 3 Report

Comments and Suggestions for Authors

The review paper submitted to cancers MDPI titled:

“Management of Hepatocellular Carcinoma in 2024: The multi-disciplinary paradigm in an evolving treatment landscape” by Emily Kinsey and Hannah M. Lee presents a concise description of the most recent and relevant clinical trials documenting progress made in treatment modalities of hepatocellular carcinoma. The paper is well organized and divided into specific treatments depending on the disease burden. Overall paper is well structured with defined chapters with proper use of references and relevant examples. However, there are some small edits, such as abbreviations used that are not completely clear, what could make this paper difficult to follow and, if not properly explained, risk loss of an intended message.

Please, carefully edit this paper prior publication, and make sure that the abbreviations used are clearly stated in the Tables legends, as well as, some abbreviations used in the main text of this paper.

Comments on the Quality of English Language

There is no comment on English language use in this paper. It is not confusing, only please remember to include in the main text, Tables and Figures and accordingly refer to the information included in the Tables, as this will help clarify the message.

Author Response

We thank the reviewer for their comments. We have attached the updated manuscript with tables 3-4 (at end of manuscript). Please see our comments in black bold below.

Review 3:

Comments and Suggestions for Authors

The review paper submitted to cancers MDPI titled:

“Management of Hepatocellular Carcinoma in 2024: The multi-disciplinary paradigm in an evolving treatment landscape” by Emily Kinsey and Hannah M. Lee presents a concise description of the most recent and relevant clinical trials documenting progress made in treatment modalities of hepatocellular carcinoma. The paper is well organized and divided into specific treatments depending on the disease burden. Overall paper is well structured with defined chapters with proper use of references and relevant examples. However, there are some small edits, such as abbreviations used that are not completely clear, what could make this paper difficult to follow and, if not properly explained, risk loss of an intended message.

Please, carefully edit this paper prior publication, and make sure that the abbreviations used are clearly stated in the Tables legends, as well as, some abbreviations used in the main text of this paper.

We have taken out the table with many abbreviations that may have been hard to understand.

Comments on the Quality of English Language

There is no comment on English language use in this paper. It is not confusing, only please remember to include in the main text, Tables and Figures and accordingly refer to the information included in the Tables, as this will help clarify the message.

 We have taken out the table with lots of abbreviations.  

Round 2

Reviewer 1 Report

Comments and Suggestions for Authors

The authors significantly improved their paper but some points still remain unanswered. I think a small mention on prevention at least in specific subsets of patients (See my previous comment about obese patients) should be added.

Author Response

We have made a mention of the study regarding bariatric surgery and HCC prevention in MASLD (lines 116-126/ pg 4)

Reviewer 2 Report

Comments and Suggestions for Authors

As TACE can be repeated (and indeeed several rounds of TACE can be performed), the Authors should mention under which circumstances "TACE failure" should be detected and sytemic therapy started.  

Author Response

We have added additional text regarding the multiple use of TACE and when to switch to other therapies  (lines 311-318/ pg 9)